# Differential Diagnosis of Preeclampsia Based on Urine Peptidome Features Revealed by High Resolution Mass Spectrometry

**DOI:** 10.3390/diagnostics10121039

**Published:** 2020-12-03

**Authors:** Alexey S. Kononikhin, Natalia V. Zakharova, Viktoria A. Sergeeva, Maria I. Indeykina, Natalia L. Starodubtseva, Anna E. Bugrova, Kamila T. Muminova, Zulfia S. Khodzhaeva, Igor A. Popov, Wenguang Shao, Patrik Pedrioli, Roman G. Shmakov, Vladimir E. Frankevich, Gennady T. Sukhikh, Evgeny N. Nikolaev

**Affiliations:** 1Skolkovo Institute of Science and Technology, 143026 Moscow, Russia; E.Nikolaev@skoltech.ru; 2V.I. Kulakov National Medical Research Center for Obstetrics, Gynecology and Perinatology, Ministry of Healthcare of the Russian Federation, 117198 Moscow, Russia; anna.bugrova@gmail.com (A.E.B.); k_muminova@oparina4.ru (K.T.M.); z_khodzhaeva@oparina4.ru (Z.S.K.); popov.ia@phystech.edu (I.A.P.); r_shmakov@oparina4.ru (R.G.S.); v_frankevich@oparina4.ru (V.E.F.); g_sukhikh@oparina4.ru (G.T.S.); 3Emanuel Institute for Biochemical Physics, Russian Academy of Sciences, 119334 Moscow, Russia; nvzakharova@ya.ru (N.V.Z.); viktoriya.shirokova@phystech.edu (V.A.S.); mariind@yandex.ru (M.I.I.); 4V. L. Talrose Institute for Energy Problems of Chemical Physics, N. N. Semenov Federal Center of Chemical Physics, Russian Academy of Sciences, 119334 Moscow, Russia; 5Moscow Institute of Physics and Technology, 141701 Moscow, Russia; 6Institute of Molecular Systems Biology, ETH Zurich, 8092 Zurich, Switzerland; wenguang.shao@imsb.biol.ethz.ch (W.S.); pedrioli@imsb.biol.ethz.ch (P.P.)

**Keywords:** urine peptidomics, preeclampsia, hypertension, proteomics, mass-spectrometry

## Abstract

Preeclampsia (PE) is a severe pregnancy complication, which may be considered as a systemic response in the second half of pregnancy to physiological failures in the first trimester, and can lead to very serious consequences for the health of the mother and fetus. Since PE is often associated with proteinuria, urine proteomic assays may represent a powerful tool for timely diagnostics and appropriate management. High resolution mass spectrometry was applied for peptidome analysis of 127 urine samples of pregnant women with various hypertensive complications: normotensive controls (*n* = 17), chronic hypertension (*n* = 16), gestational hypertension (*n* = 15), mild PE (*n* = 25), severe PE (*n* = 25), and 29 patients with complicated diagnoses. Analysis revealed 3869 peptides, which mostly belong to 116 groups with overlapping sequences. A panel of 22 marker peptide groups reliably differentiating PE was created by multivariate statistics, and included 15 collagen groups (from COL1A1, COL3A1, COL2A1, COL4A4, COL5A1, and COL8A1), and single loci from alpha-1-antitrypsin, fibrinogen, membrane-associated progesterone receptor component 1, insulin, EMI domain-containing protein 1, lysine-specific demethylase 6B, and alpha-2-HS-glycoprotein each. ROC analysis of the created model resulted in 88% sensitivity, 96.8% specificity, and receiver operating characteristic curve (AUC) = 0.947. Obtained results confirm the high diagnostic potential of urinary peptidome profiling for pregnancy hypertensive disorders diagnostics.

## 1. Introduction

Preeclampsia (PE) is the most severe hypertensive pathology complicating 2–8% of pregnancies and is associated with increased risk of miscarriage, premature birth, disability in the newborns, and development of severe cardiovascular pathologies in women after pregnancy, as well as neonatal and maternal deaths [1,2,3,4]. Unlike other hypertensive pregnancy complications (chronic or gestational hypertension), PE occurs after the 20th week of pregnancy and is often associated with proteinuria and other signs of multiple organ dysfunction [5,6,7,8,9,10]. Timely and appropriate treatment is of particular significance to avoid the fateful outcomes. The identification of women at high risk before pregnancy or at least until the 13th week of gestation is extremely important for choosing an adequate management as abnormal placentation leads to more severe forms of the pathology and its early onset [11]. However, late onset PE (≥34 weeks’ gestation) may occur due to other intrinsic pathologies that may be triggered by pregnancy and are also related with abnormal uteroplacental and vascular remodeling, as well as with redistribution of blood flow, which is laid in the first trimester of pregnancy [12,13,14,15].

Several physiological pathways related to genetic, epigenetic, and environmental factors may be involved in PE [16]. Main risk factors include primipregnancy, long interval between pregnancies, advanced maternal age, pre-pregnancy obesity, diabetes, hypertension, antiphospholipid syndrome, ethnic background, and a previous and/or family history of PE [11,17,18,19]. The etiology of PE essentially relates to hypoxia resulting from abnormal trophoblast invasion and vascular and uterine remodeling. Predisposing genetic factors relate to polymorphisms in a number of proteins/genes essential for regulation of blood coagulation, vascular endothelial function, blood pressure, inflammation, and immunity, which are also essential in other systemic pathologies not associated with pregnancy, such as hypertension, vascular disease, thrombophilia, and systemic inflammation [19,20,21,22,23,24]. The dysregulation in expression of SERPIN proteins A3, A5, A8, B2, E1, E2, and G1 shown earlier may reflect some compensatory mechanisms in PE-placentas [16]. Thus, the maternal PE syndrome that appears in the second half of pregnancy can be considered as a systemic response to systemic failures in the first trimester, which can occur under various scenarios, depending on different combinations of numerous genetic and non-genetic factors, and may not depend only on a few of them.

Since the real causes of PE are laid long before its manifestation, there is an opportunity for timely diagnostics and appropriate management to improve maternal and perinatal outcomes. However, reliable screening tests for clinical application have not yet been developed. Among protein markers, abundant expression of metalloproteinases MMP-2 and MMP-9 (gelatinases A and B) in extravillous trophoblasts is highly related to extracellular matrix (ECM) degradation [25,26,27]. Levels of MMP-2 and its inhibitor TIMP-1 were shown to be significantly increased in PE [28,29]. A decreased circulation level of placental growth factor (PLGF) may predict the development of PE in the following 2 weeks [30,31]. Soluble fms-like tyrosine kinase-1 (sFlt-1) can block vascular endothelial growth factor (VEGF) and PLGF via their binding. Its increased level (or increased sFlt-1/PLGF ratio) may precede PE from the second trimester onwards [31,32,33]. The increased degree of circulating anti-angiogenic soluble endoglin (sEng) strongly correlates with PE severity [34,35]. Decreased levels of placental protein-13 and pregnancy-associated protein A in the first trimester as well as increased levels of pro-inflammatory cytokines and hypoxia induced factor 1α may also suggest PE; however, these features are less specific [36,37,38,39,40].

Urine seems to be the most convenient subject for research due to its relative stability and non-invasive collection. Since PE is often associated with renal pathologies and proteinuria, urine proteomic analysis may provide valuable information, for example it may distinguish PE from other hypertensive pregnancy complications, and may estimate the degree of disease severity, which is essential for further management. Mass spectrometry (MS)-based approaches proved to be the most effective in previous urine peptidome/proteome studies and provided most of the current information [41,42,43,44,45,46,47]. Currently, the proposed urine PE markers include reduced levels of PlGF, prostaglandin-H2 D-isomerase, and perlecan [41,48], as well as increased levels of specific fragments of α-1-antitrypsin (SERPINA1), albumin, fibrinogen alpha chain, collagen alpha chain, uromodulin [41,42,43,44,45], and of some unidentified proteins [46,47]. Using high-performance liquid chromatography with tandem mass spectrometry (HPLC-MS/MS), we found 35 specific urine peptides originating from SERPINA1, uromodulin, and collagen alpha-1 chains (I and III), which reliably distinguished a particular PE group (10—mild PE; 10—severe PE) from normal controls [45]. However, the results of various studies are not consistent enough, and none of the particular markers described to date show sufficient sensitivity. We believe that creation of a differentiating peptide marker panel can essentially improve PE diagnostics, and remains of particular importance as a basis for the further development of accessible clinical methods.

Here, based on the analysis of 127 urine samples of pregnant women (including normotesive controls, chronic hypertension (CH), gestational hypertension (GH), moderate PE (mPE), and severe PE (sPE)), and using peptide grouping, we propose a variant of such peptide panel with a high differentiating capacity, which consists of 22 peptide groups and also includes markers described earlier.

## 2. Materials and Methods

### 2.1. Patients

Patients were diagnosed in V.I. Kulakov’s National Medical Research Center of Obstetrics, Gynecology and Perinatology in accordance with the ACOG Practice Bulletin (2002). PE was diagnosed at blood pressure (BP) > 140/90 mmHg, proteinuria > 0.3 g/L daily, edema, manifestations of multisystem organ insufficiency. Severe PE was diagnosed with at least one of the following symptoms: BP ≥ 160/100 mmHg (twice within 6 h); proteinuria ≥ 5 g/L daily or > 0.3 g/L in separate urine samples; oliguria ≤ 500 mL/day; epigastric pain and/or pain in right hypochondrium; pulmonary edema or pulmonary failure; more than twice increase in alanine aminotransferase (ALT) and aspartate aminotransferase (AST); neurological complications; thrombocytopenia (less than 100 × 10^9^/L); intrauterine growth restriction (IUGR) (fetal weight below 10 percentiles).

Normotensive pregnant women with urine protein content < 0.1 mg/mL were enrolled into the control group. The exclusion criteria included: multiple pregnancy, pregnancy after assisted reproductive technology (ART), diabetes, transplanted organs, autoimmune diseases, oncological diseases, decompensated kidney disease, chromosomal abnormalities in the fetus, congenital malformations of the fetus, and antenatal fetal death. CH and GH were diagnosed in patients who did not meet the PE criteria: CH in patients, who had hypertension before pregnancy; GH in patients with hypertension experienced during pregnancy.

In total, the study included 127 urine samples from 126 pregnant women: 17 from normotensive pregnant controls, 16 from patients with chronic arterial hypertension (CH; two of the samples were obtained from one patient), 15 with gestational arterial hypertension (GH), 25 with mild PE (mPE), 25 with severe PE (sPE), and 29 samples were from patients with complicated diagnoses that were considered to be unrelated to any group. The latter included 14 samples from patients with PE superimposed on CH (PE-CH); 4 samples from patients diagnosed as CH with suspected PE; 1 sample from the patient with GH who later developed mPE and sPE; 1 sample with mPE superimposed on a non-hypertensive pathology; and 9 samples with suspected but not diagnosed PE. Clinical and demographical data are shown in Table 1.

In the intergroup analysis incidence of hypertension in family history in PE, CH and PE-CH did not differ statistically between the groups—46.4%, 50%, and 50%, correspondingly—but was significantly more common as compared to the control group: 20% (*p* < 0.01). Thus, hypertension in relatives is a risk factor for hypertensive disorders in pregnancy. In the PE group, women were significantly more likely to have had adverse pregnancy outcomes: antenatal fetal death (7.1%), early neonatal death (3.6%), and history of preeclampsia (17.9%) compared to women with PE superimposed on CH (0%, 0%, and 12.5%, respectively), and to the control group where these complications were not observed (*p* < 0.05).

### 2.2. Urine Sample Collection and Peptide Isolation

Urine collection was performed before treatment after a written informed consent of participants in accordance with the protocol approved by the Ethical Committee (Record No12 from 17 November 2016) of V.I. Kulakov’s National Medical Research Center of Obstetrics, Gynecology and Perinatology.

Urine samples were centrifuged (2000 g, 10 min, 4 °C) within 20 min after collection, and the supernatant was stored at −80 °C. The peptide fraction was obtained as described earlier [45,49]. Particularly, 1.5 mL of urine was diluted with 3 volumes of denaturing buffer (4M urea, 20 mM ammonium hydroxide, 0.2% sodium dodecyl sulfate), transferred to Vivaspin-4 10 kDa MWCO (Sartorious) filters and centrifuged at 4000 g for 20 min at room temperature. The filtrate (2.5 mL) was further subjected to gel-filtration on a PD-10 Column (GE Healthcare; Sephadex™ G-25 Medium, equilibration and elution with 0.01% ammonium hydroxide). An amount of 2 mL of the eluate was lyophilized and dissolved in 100 µl of deionized water prior to analysis.

### 2.3. HPLC-MS/MS Analysis

HPLC-MS/MS analysis was performed on a nano-HPLC Agilent 1100 system (Agilent Technologies, Santa Clara, CA, USA) using a homemade capillary column (id 75 μm × length 12 cm, Reprosil-Pur Basic C18, 3 μm, 100 A; Dr. Maisch HPLC GmbH, Ammerbuch-Entringen, Germany) in combination with a 7-Tesla LTQ-FT Ultra mass spectrometer (Thermo Electron, Bremen, Germany) equipped with an in-house nanospray ion source. Gradient chromatography was implemented by changing the relative concentration of solvent B (100% acetonitrile/0.1% formic acid) in flow of solvent A (0.1% formic acid). Main elution time was 15–45 min: linear gradient from 3% to 50% of solvent B, elution of most hydrophobic peptides was 45–50 min: linear gradient from 50% to 90% of solvent B. Each sample was analyzed in 4 runs.

### 2.4. Urinary Proteome Data Base Development

To facilitate the search procedure, a small data base was created for identification and semiquantitative analysis of the massive HPLC-MS/MS data (see below) using available computing power. The detailed description of the urinary proteome data base development is out of the scope of this manuscript. Briefly, to develop the data base, proteome analysis was performed for three pooled urine samples and their 10 fractions prepared by Isoelectic Focusing Electrophoresis (IEF) using the PROTEAN system (Bio-Rad). Pooled samples were prepared from 60 individual samples (10-Control, 10-GH, 10-CH, 10-(PE-CH), 10-mPE, 10-sPE) which were used for proteome and peptidome analysis. All samples were measured on a Q-Exactive HF operating in data dependent (DDA) mode. In total, 8423 tryptic peptides from 1029 protein groups were identified and archived into the library. In an attempt to increase the coverage of the library, additional 95 individual samples (from 127 patients-2.1. Patients) were selected for additional proteomic analysis and measured in data independent (DIA) mode. Overall, this resulted in a spectral library containing 11,131 peptide ions and 1472 proteins, 32% and 43% more than the initial DDA library, respectively. Finally, the database was expanded with preliminary peptide search data and peptides described previously in PE [42,43,44,45,46,47].

### 2.5. Data Analysis

Urinary peptides were identified using PEAKS Studio 8.5 and MaxQuant (version 1.6.7.0) search programs across the developed urinary proteome data base (see above) with the following parameters: non-specific enzyme; mass accuracy for the precursor ion-20ppm; mass accuracy for MS/MS fragments—0.50 Da; possible variable modification–oxidation of methionine, lysine, and proline residues: up to 5 variable modifications per peptide, minimal peptide length was set to 5 amino acid residues, maximal peptide weight—10 kDa, false discovery rate (FDR) ≤ 0.01, minimal score for unmodified peptides—30, for modified—40. For semiquantitative label-free analysis, alignment of chromatograms was used; the particular peptide peak intensity values were normalized to the total intensity of all peaks in a particular sample. Significant differences in the representation of peptides in different patient groups were estimated using the Mann–Whitney U-test. The Venn diagram was build using the http://bioinformatics.psb.ugent.be/webtools/Venn/ resource, and heat-maps were created with the http://heatmapper.ca/ resource [50].

Multivariate data analysis of the semiquantitative proteomic data was performed using partial least squares analysis (PLS) with the ropls package [51] to create a classification model: normotensive pregnancy (control), hypertension without PE (CH and GH) and PE (mild and severe). The Y variables for PLS model training were set from 1 to 3 for samples of different groups (specified in Figures). The quality of statistical models was estimated by R2 (fraction of data that the model can explain using the latent variables) and Q2 (fraction of data predicted by the model according to the cross validation) parameters. The model performance was assessed by calculating the area under receiver operating characteristic curve (AUC).

## 3. Results

### 3.1. Urine Peptidome Analysis

Analysis of MS data obtained for 127 samples allowed the identification of a total of 3869 peptides with FDR < 0.1% from 43 proteins when using the PEAKS program for identification (Appendix A). However, a much smaller number of peptides was found to be substantially represented in at least one of the diagnostic groups: calculation of the median intensity values revealed only 340 (including PTM variations) substantially represented peptides, whose medians exceeded the ”zero” level (Table 2 and Appendix A).

Data in Table 1 indicates that the vast majority of substantially represented peptides are derived from collagen alpha-1(I) (COL1A1) and alpha-1(III) (COL3A1) chains. Single peptides from membrane-associated progesterone receptor (PGRMC1), neurosecretory protein VGF, insulin (INS), and lysine-specific demethylase 6B (KDM6B) are of particular interest as they have a predominant distribution in the hypertensive or PE groups. It is important to note that MaxQuant results are mostly consistent with those of PEAKS (Table 2, Appendix A); however, PEAKS de novo sequencing is a definite advantage that essentially expands its identification capabilities. So, further analysis was mainly performed on the PEAKS data.

Unexpectedly, none of the SERPINA1 peptides demonstrated a substantial presence in any of the groups, although to date, their presence in urine of pregnant women has been considered as the main PE marker [42,45]. Nevertheless, the total number of identified SERPINA1 peptides (506 of 3869) is in third place after COL1A1 and COL3A1 ones (1493 and 917, correspondingly, Appendix A). This suggests that SERPINA1 peptides are actually substantially represented, but their excision sites are highly variable. At the same time, the mostly represented SERPINA1 peptides (up to 37 samples, Appendix A) were found to correspond to the C-terminus and have overlapping sequences. Therefore, it was suggested that combining peptides with overlapping sequences could take SERPINA1 peptides into consideration, as well as essentially expand the number of differentiating peptides. For this, we further considered peptide groups and compared the summed intensities of overlapping peptides.

### 3.2. Peptide Groups

The initial 3869 peptides (PEAKS) were subdivided into 116 groups (loci) in accordance with the overlapping sequences (Appendix A), and 62 of these groups were found to be substantially represented in at least one patient group (Figure 1, Table 3). Interestingly, these substantial groups mostly originate from the proteins mentioned in Table 2, and only one COL5A1 and one SERPINA1 (C-terminal) loci were added. The number of samples, in which a particular peptide group was identified, increased in comparison with corresponding individual peptides (Table 3). It seems very important that the peptide group distribution mainly coincides with the distribution of individual essentially represented peptides, and thus suggests that consideration of peptides with overlapping sequences as one group is quite reasonable, especially since it allows information from a significantly higher number of peptides to be taken into account.

The data in Table 3 indicate that the bulk of the peptide loci is represented in all patient groups and belongs to the core, which mostly includes collagen (COL1A1, COL3A1, and others), KISS1, EMID1, fibrinogen, and uromodulin peptides. The heat-map in Figure 1B, however, suggests the essential differences in the percentage of these core loci in samples with different diagnoses. Single peptide groups from PGRMC1, VGF, insulin, and COL5A2 are substantially represented in CH and GH samples. The group of KDM6B peptides is substantially present in PE samples, like SERPINA1 peptides. It is noteworthy that consideration of overlapping peptides together as a single group essentially increased the number of samples with accounted KDM6B peptides (from 42 to 62), and reinforced the indication that these peptides could be potential PE markers.

Pairwise data comparison for all sample groups using the Mann–Whitney U-test revealed a significantly different distribution for 52 of the 62 peptide loci (Appendix A). No significant difference was shown for 4 loci from COL3A1, 3 of COL1A2, as well as single EMID1, FXYD2, and uromodulin groups. Among the significantly differentiating peptide groups, 17 seem to be especially characteristic for the control, hypertensive, or PE samples, according to their *p*-values (Appendix A, Figure 2).

The clustering in Figure 2 shows a good trend in the distribution of samples in accordance with the clinical diagnoses. This suggests that these urinary peptide groups can be considered as an appropriate basis for further development of a PE diagnostic panel.

### 3.3. Predictive Performance and Model for PE Differentiation

The PLS method was further applied to build a statistical model differentiating PE from hypertensive and control groups. In the beginning, the set of 570 individual peptides represented in at least 15 samples (Appendix A) was analyzed to estimate whether some of potentially differentiating peptides could be lost upon selection of substantially represented peptides (and their groups). The clustering of three sample groups (control versus CH+GH versus PE) on these 570 peptides revealed 103 VIP-peptides, which give the largest contribution to the projection on hidden structures (Appendix A), of which only the peptide 323–339 from alpha-2-HS-glycoprotein (AHSG) did not fall into consideration in the field of view above. Therefore, its peptide group was also taken into account and further PLS analysis was performed for 63 peptide loci (Appendix A). Multiple clustering with different combinations of samples revealed the best parameters for combinations ”Control+CH+GH versus mPE versus sPE” and ”CH+GH versus mPE versus sPE” (Figure 3A,B). The comparison of VIP-peptide groups obtained upon multiple clustering showed that 22 of them were most often selected as VIPs upon clustering of different combinations of samples (Table 4). They included 15 different collagen loci as well as single groups from SERPINA1, KDM6B, INS, PGRMC1, EMID1, FGA, and AHSG; 13 of them were also part of the set of 17 selected in the previous section (Figure 2). Clustering by these 22 peptide loci showed the best parameters for differentiating mPE and sPE from ”Control+CH+GH”, while the best parameters for PE differentiation from hypertensive samples only (CH+GH) was obtained with only 20 of these groups (excluding loci ”540–573” and ”798–812” of COL1A1, Table 4).

### 3.4. PE Markers

SERPINA1-derived peptides are currently one of the main commonly accepted urine PE markers [42,45,47,52,53]. It is noteworthy that among the 3869 peptides identified in this study, 506 derived from SERPINA1. In accordance with their sequences, they can be grouped into 11 loci (Appendix A) and cover 64.8% of the full sequence. Although the C-terminal locus (397–418) was shown to be the only one substantially represented in PE-samples, C-terminal peptides were also found in 2 CH samples and 16 samples with complicated diagnoses. From the 50 PE-samples, these C-terminal peptides were found in 28 (14 mPE and 14 sPE), and only 2 PE samples contained other SERPINA1 peptides while missing the C-terminal locus. At the same time, peptides from other SERPINA1 groups though each was met in only a small number of samples, were found in PE samples only, thus suggesting that identification of all SERPINA1 peptides may still remain reasonable for PE diagnosis. However, 20 of the PE samples (40%) did not contain any SERPINA1 peptides and required other diagnostic marker(s).

The KDM6B-derived poly-proline peptides 252–257/263 may be another PE-marker as they show predominant significantly different distribution in PE patient groups (in 16 mPE and 17 sPE samples). The data in Figure 4 implicate that together SERPINA1 and KDM6B peptides identify 47 of the 50 PE samples. However, KDM6B peptides demonstrated lower specificity in comparison with SERPINA1 and were also found in 25% normotensive and CH samples and in 46.7% GH samples. Nevertheless, the poly-proline peptide group is the essential integral component of the described above differentiating panel.

The AHSG peptide group 321–339 is one another possible PE marker. Although it is much less represented, only in 8 mPE and 10 sPE samples, it seems to be rather specific. However, these peptides were found only in samples, which also contained SERPINA1 or poly-proline KDM6B peptides, and did not increase the total number of identified PE samples (Figure 4). Nevertheless, this peptide group also deserves particular attention due to its selection into PLS-VIPs (Table 4).

### 3.5. Samples from Patients with Complicated Diagnoses

MS data for 29 ungrouped samples were analyzed both for the presence of PE markers (peptides from SERPINA1, KDM6B, and AHSG) and for the characteristic clustering of VIP-peptide loci identified above. When diagnosed by markers, based on the data obtained for each patient group, the presence of SERPINA1 or AHSG peptides most likely indicated PE; sPE was highly probable if normalized intensity values for SERPINA1 C-terminal peptide locus exceeded ”3”. The KDM6B poly-proline peptide group could suggest PE if its normalized intensity values were higher than ”0.1”. For diagnosis with VIP-peptide groups, the data of ungrouped samples were co-clustered with the data of the grouped samples, which were used above for the identification of VIP-peptides. The particular co-clustering suggested the most likely diagnosis (Figure 5).

According to the results in Figure 5, VIP-peptide co-clustering is essentially more sensitive and specific than diagnosis by markers, since it gives 90% coincidence with clinical diagnoses and is able to differentiate any sample, whereas the markers give only a 67% coincidence and differentiate only 20 of the 29 samples.

The sample from the patient, who was clinically diagnosed as GH and further developed mPE and sPE (number 6*, green arrow with asterisk in Figure 5), is of particular interest, since the two later samples were also analyzed (and enrolled into mPE and sPE diagnostic groups). The illustrated in Figure 6 dynamic changes of the peptide profile and VIP-peptide groups for this patient suggest that in addition to a significant increase in PE markers, the significant decrease in the content of collagen, INS, PGRMC1, and other peptides has a quite pronounced tendency and may reflect PE severity as well as appearance in the profile of a large number of peptides originating from a variety of plasma proteins (such as APOA1, A1BG, HBB, and others).

Importantly, SERPINA1 peptides were found in all three samples of this (initially GH) patient, although their content was very significantly different (Appendix A). The fact that VIP-peptide clustering still differentiated the first sample as GH in spite of the presence of SERPINA1 peptides (what coincided to the clinical diagnosis) once again implicates the high diagnostic capacity of the obtained VIP-panel. At the same time, in the example of this patient, it can be assumed that the appearance of SERPINA1 peptides in urine may indicate the increased risk of GH progression to PE.

In general, the results obtained with ungrouped samples from patients with complicated diagnosis suggest the essential diagnostic potential of the obtained VIP-peptide panel.

## 4. Discussion

Creation of a urine peptide panel for PE differentiation remains an urgent research task. In an attempt to create such a panel, a special strategy was applied in this study. Urine peptides with overlapping sequences were grouped and considered together as one locus and were found to have the same distribution as their most represented peptides. The combination of peptides seemed appropriate given that no specific cleavage sites and amino-acid modifications were detected for any diagnosis, at least in the analyzed 127 samples. It was assumed that peptide grouping and comparison of their summed normalized intensities should smooth out the insignificant differences of sequences and focus the attention on the presence of particular peptide locus and its relative content in samples with specific diagnoses. Additionally, this strategy might essentially decrease the discrepancies when measuring different series of samples and even reduce dissimilarities in the results of different studies. The obtained results suggested that the application of this strategy proved to be quite appropriate and led to the creation of a variant of a peptide panel, which differentiates PE with high sensitivity and specificity.

Among members of the obtained panel there are several coincidences with the results of other studies (Table 4). In particular, two COL1A1 groups, ”510–539” and ”765–794”, and the FGA ”604–624” locus contain peptides earlier described as PE markers by Carty D.M. et al. [44]. Among other substantially represented groups obtained here, COL1A1 ”1007-1041”; COL3A1 ”446-477”, COL1A2 ”451–472”, and ”917–952”; and the uromodulin locus also have coincidences with the results of [44]. However, here, they were not selected into VIP-groups, and moreover, the distribution of uromodulin peptides was not found to be significantly different (remarkably, all of the revealed uromodulin peptides belonged to the same locus ”589–607”). It is noteworthy that all of these peptide groups (including 3 VIPs) were identified in most of the analyzed samples and, hence, their presence by itself is not a marker. Nevertheless, their relative content (and in particular the content of VIP-groups) proved to be extremely important for PE differentiation.

As for direct markers of PE, the obtained results support the findings of I.A. Buhimschi et al. concerning SERPINA1 peptides [42]. However, despite the wide variety of identified SERPINA1 peptides, only the C-terminal group was shown to have a diagnostic significance since it had the highest representation. Again, consideration of the C-terminal peptide locus ”397–418” also proved to be advantageous compared to its most represented peptide (398–418) since the group was detected in 47 different samples, while the peptide itself in only 37 of them. However, SERPINA1 peptides were found in only 60% of PE samples and the rest required other markers.

Poly-proline (from 6 to 12 poly-P) peptides, SERPINA1 peptides, were found to be significantly distributed in PE patient groups and essentially enhanced PE differentiation; together with SERPINA1 peptides, poly-P covered 94% of PE samples. Unlike SERPINA1 peptides, poly-P peptides are not absolutely specific markers; nevertheless, their relative content is definitely an essential indicator, first identified in this study. The possible relation of KDM6B to PE is actually unclear, especially taking into account that this protein is associated with chromatin and localized in the nucleus. However, it was shown to be involved in the inflammatory response by participating in macrophage differentiation via the regulation of gene expression [54]. Since PE is usually associated with a systemic inflammation, it may be assumed that KDM6B peptides in urine may originate from the degraded macrophages. On the other hand, the appearance of KDM6B peptides may reflect the epigenetic regulation of compensatory mechanisms, which, in particular, may lead to changes in the expression level of different SERPINs [16]. In addition, poly-P peptides may actually have different origin. In particular, many cytosolic, membrane, and cytoskeletal proteins such as large proline-rich protein BAG6, WASP homolog-associated protein with actin, protein diaphanous homologm1, MAP3K4 kinase, Ras-associated and pleckstrin homology domains-containing protein 1, and junction-mediating and -regulatory protein contain poly-P motives and may be the source of poly-P peptides in cell degradation. Still, the mechanism underlying the increase in urine poly-P in PE remains questionable.

Among the PE markers identified in this study, AHSG seems to be the most doubtful, since it was found only in 26 of 127 samples. Additionally, there are several other peptide groups with similar representativeness that did not fall into the differentiating panel: apolipoprotein A1 ”254–267” (28 mostly PE samples), complement C4-A ”1423–1440” (27 samples except for controls), clusterin ”390–423” (39 samples except for controls) and some other (Appendix A). In general, it is important to note that sPE is often associated with a large variety of urine peptides originating from different plasma proteins, and these peptides may look highly PE-specific. Whether it is appropriate to consider all of such peptides as possible markers remains a question.

In sum, it is worth noting that the obtained differentiating panel for the most part consists of peptides that are represented in most samples of all diagnostic groups. The relative content of these peptides implicates the important criterion for diagnosis that is similar to the presence of specific markers. The panel still requires further validation and may be essentially reorganized. However, the differentiating role of the substantially represented peptides and their groups is expected to be the most promising.

## 5. Conclusions

High-resolution mass spectrometry applied for analysis of the peptidome of 127 patients including normotensive controls (*n* = 17), chronic hypertension (*n* = 16), gestational hypertension (*n* = 15), mild PE (*n* = 25), severe PE (*n* = 25), and 29 patients with complicated diagnosis reveal sample diversity and new features for PE diagnostics. A panel consisting of 22 peptide loci from collagens (COL1A1, COL3A1, COL2A1, COL5A1, COL8A1, and COL4A4), fibrinogen alpha-chain, insulin, membrane-associated progesterone receptor component 1, EMI domain-containing protein 1, alpha-1-antitrypsin, lysine-specific demethylase 6B, and alpha-2-HS-glycoprotein was developed. Reliable differentiation of preeclampsia from chronic or gestation hypertension and from normotensive cases was demonstrated with 88% sensitivity and 96.8% specificity (AUC = 0.947). Overall, the obtained results confirm the high diagnostic potential of urinary peptidome profiling and can serve as the basis for further creation of new reliable methods for clinical diagnostics of preeclampsia.

## Figures and Tables

**Figure 1 diagnostics-10-01039-f001:**
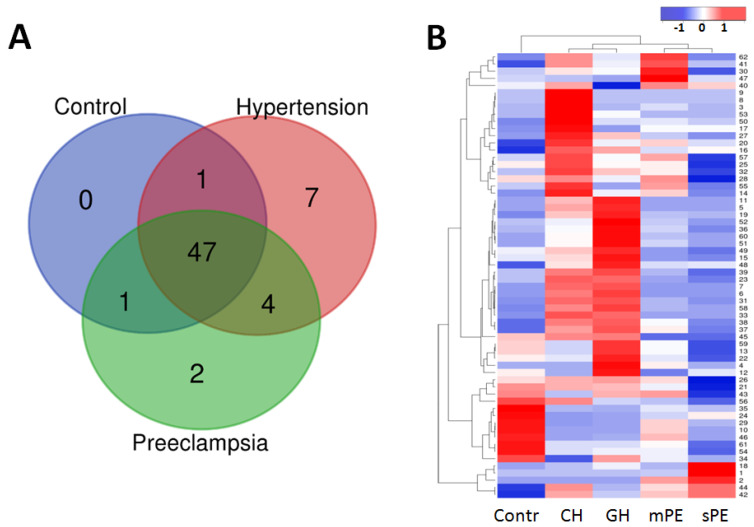
The distribution of 62 substantially represented peptide loci in control, hypertension (CH and GH), and preeclampsia (mPE and sPE) patient groups. (**A**) Venn diagram demonstrating peptide group intersections. (**B**) Hierarchical clustering of the peptide loci’s median intensity values and pregnancy associated hypertensive disorders groups. The Kendal’s Tau distance measurement method and average linkage clustering were used. The higher values are shown in red, the lower—in blue.

**Figure 2 diagnostics-10-01039-f002:**
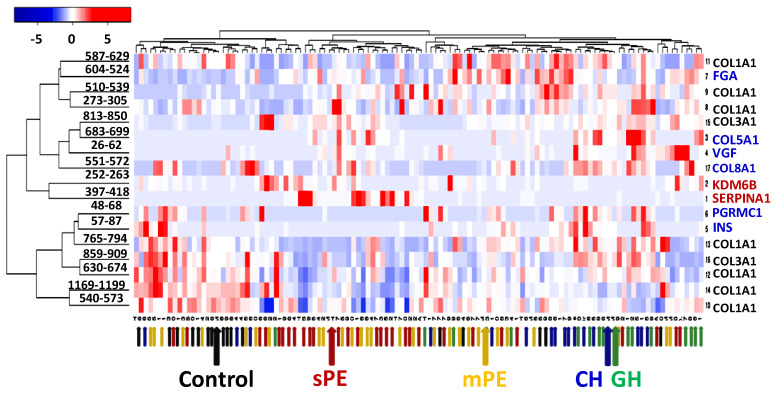
Hierarchical clustering of individual sample data and 17 characteristic peptide groups using Kendal’s Tau distance measurement method and average linkage clustering. The small colored arrows indicate the clinical diagnosis in each particular sample; the large arrows show the median data of the corresponding patient groups: black—control, blue—CH, green—GH, yellow—mPE, and red—sPE.

**Figure 3 diagnostics-10-01039-f003:**
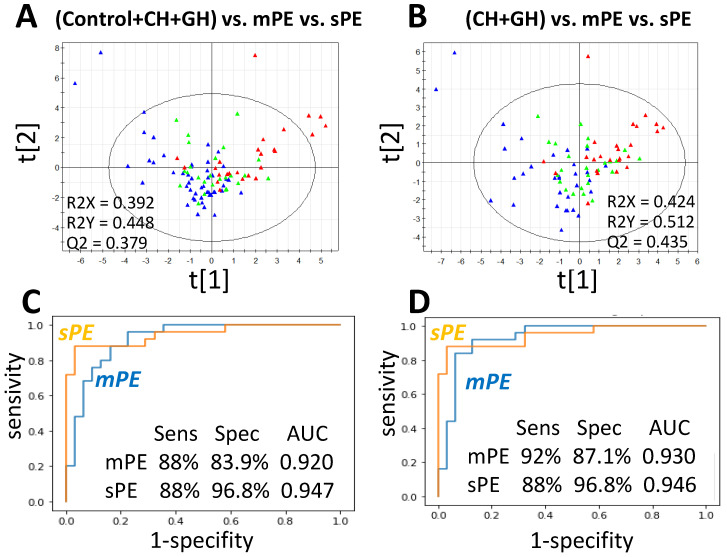
PLS model differentiating mPE and sPE urine samples from control, CH, and GH samples. (**A**,**B**) PLS score plot of semiquantitative urinary peptidomic data: blue—”Control+CH+GH” (**A**) or ”CH+GH” (**B**); green—mPE; red—sPE. (**C**,**D**) ROC analysis for mPE or sPE versus ”Control+CH+GH” (**C**) or ”CH+GH” (**D**) according to the results of clustering on 22 and 20 VIP-peptide groups (for **C** and **D**, correspondingly).

**Figure 4 diagnostics-10-01039-f004:**
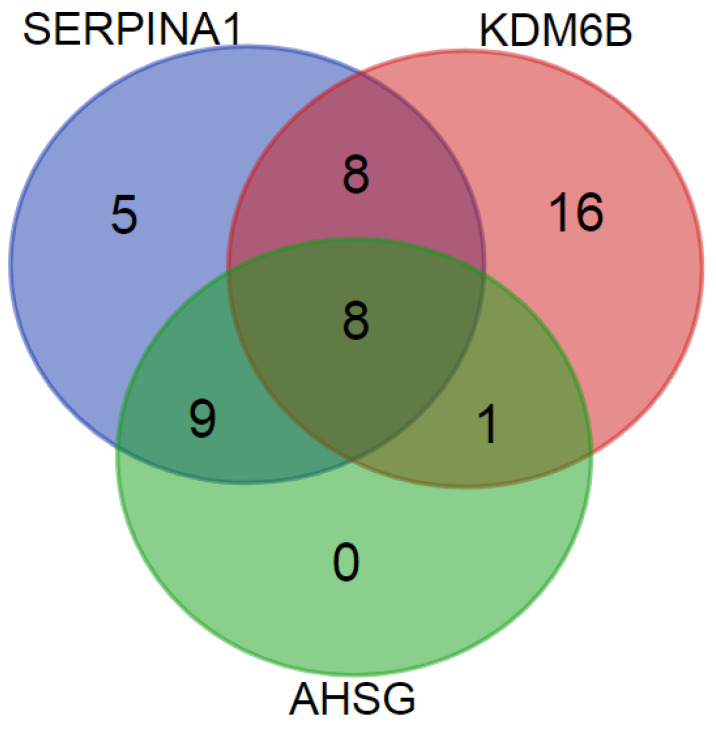
Distribution of SERPINA1, KDM6B, and AHSG peptides among 50 PE urine samples.

**Figure 5 diagnostics-10-01039-f005:**
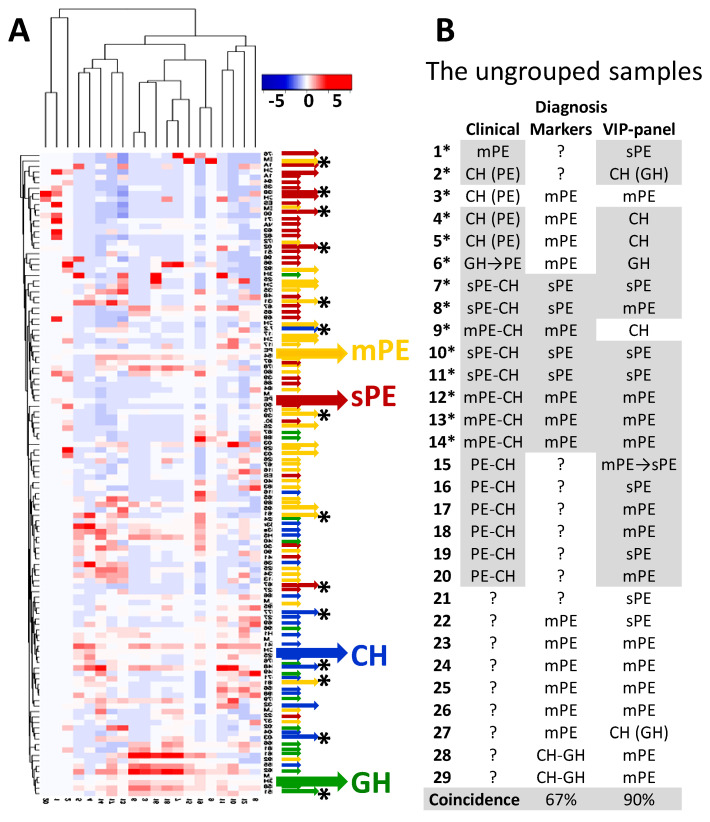
Co-clustering of ungrouped and grouped samples by 20 VIP-peptide groups. (**A**) Hierarchical co-clustering using Kendal’s Tau distance measurement method and average linkage clustering. The small colored arrows indicate the established clinical diagnoses in grouped samples; the middle arrows without asterisks—the probable diagnoses of ungrouped samples (with asterisks —established clinical diagnoses for ungrouped samples); the large arrows show the mean data of corresponding patient groups: blue—CH, green—GH, yellow—mPE, and red—sPE. (**B**) The comparison of clinical diagnoses with diagnoses proposed by using markers and VIP-peptide clustering. Grey background shows results consistent with clinical diagnoses (ignoring PE subdivision).

**Figure 6 diagnostics-10-01039-f006:**
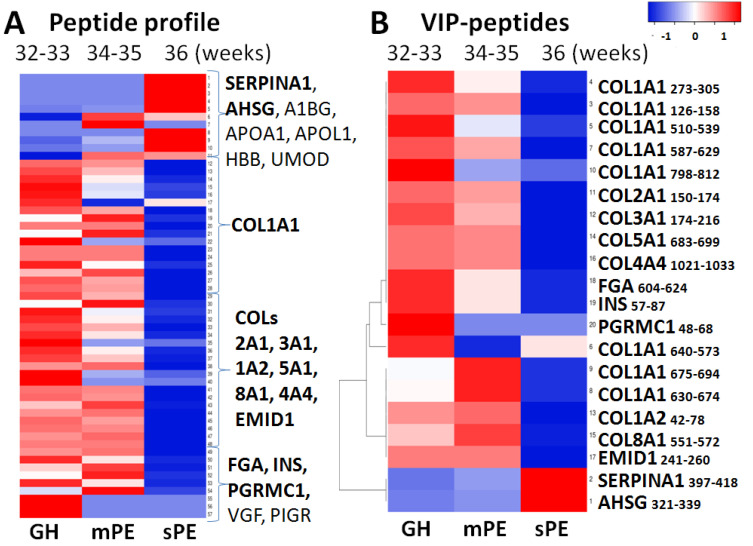
Dynamic urine peptidome changes upon GH progression into PE, on the example of three samples obtained from the same patient at different gestation ages. (**A**) Profiling of the most represented peptide groups (without clustering). (**B**) Hierarchical clustering of VIP-peptide groups found in the compared samples using Kendal’s Tau distance measurement method and average linkage clustering.

**Table 1 diagnostics-10-01039-t001:** Clinical and demographic data of patients.

Parameter	Control (*n* = 17)	CAH (*n* = 15)	GAH (*n* = 15)	Mild PE (*n* = 25)	Severe PE (*n* = 25)
Age (years)	30.5 ± 4.1	31.8 ± 5	32.6 ± 4.2	28.5 ± 4.2	32.2 ± 4.5
Height (cm)	168.6 ± 4.8	167.4 ± 5	166.8 ± 7.1	164.6 ± 5.2	163.9 ± 5.8
Weight (kg)	70.1 ± 8.3	83 ± 15.5	76.5 ± 10.3	76.1 ± 14.9	74.2 ± 14.2
BMI (kg/m^2^)	24.7 ± 3.1	28.6 ± 3.4	27.2 ± 3.1	28.1 ± 5	27.6 ± 5.1
Kidney disease	2 (11.1%)	2 (10.5%)	4 (25%)	2 (7.7%)	4 (15.4%)
Previous PE	1 (5.6%)	6 (31.6%)	1 (6.2%)	0 (0%)	6 (23.1%)
Primiparous	5 (27.8%)	7 (36.8%)	5 (31.2%)	16 (61.5%)	11 (42.3%)
Start of hypertension (days)	-	28 ± 9.1	29.2 ± 5.4	34.3 ± 5.1	28.2 ± 8.2
Start of proteinuria (days)	-	36.5 ± 0.7	33.5 ± 0.2	36.7 ± 2.1	32 ± 4.6
sFlt-1/PLGF	6.7 ± 1.8	33.4 ± 31.1	67.8 ± 78.1	229.8 ± 209.6	346.3 ± 281.9
Delivery (weeks)	39 ± 1.1	37.8 ± 1.9	37.6 ± 2.2	37.9 ± 1.6	33.6 ± 4.4
Maximal SP	117.2 ± 5.5	143.1 ± 14.4	146.7 ± 10.9	151.5 ± 12.6	157.7 ± 11.3
Maximal DP	77.8 ± 3.9	90.9 ± 7.3	94.7 ± 8.9	96.0 ± 6.9	100.8 ± 7.5
Maximal Pu (g/l)	0 ± 0	0.1 ± 0.1	0.1 ± 0.1	1.2 ± 1.0	2.3 ± 1.6
LDH	393.3 ± 0	327.6 ± 14.4	366.2 ± 33.8	334.5 ± 91.0	445.8 ± 117.1
ALT	13.8 ± 2.2	15 ± 6.6	17.9 ± 7	24.6 ± 28.7	36.1 ± 20.1
AST	17.3 ± 3.4	18.9 ± 4.6	16.8 ± 4.8	25.0 ± 12.2	33.9 ± 15.7
ALP	129.7 ± 3.4	201 ± 156.3	164.3 ± 68.3	168.5 ± 47	183.2 ± 89.2
Platelet count	217.3 ± 56.5	244.1 ± 73.4	251.5 ± 55.2	218.8 ± 67.8	199.5 ± 77.2
UAPI	0.9 ± 0	0.9 ± 0.3	1 ± 0.3	1.4 ± 0.6	2.0 ± 1.0
UMAPI	0.9 ± 0	1.1 ± 0.4	1 ± 0.3	1.3 ± 0.5	1.6 ± 1.3
MCAPI	1.6 ± 0	2.8 ± 0.7	1.8 ± 0.3	2.6 ± 1.2	2.8 ± 1.1
IUGR	0 (0%)	0 (0%)	2 (12.5%)	6 (23.1%)	17 (65.4%)
Child weight (g)	3361.8 ± 505.7	3035.3 ± 518.6	2961 ± 562.5	2768.3 ± 540.5	1783.3 ± 834.4
Child height (cm)	51.4 ± 2.4	50.5 ± 2.6	49.9 ± 3.6	48.8 ± 2.6	40.9 ± 8
Apgar 1 min	8.1 ± 0.3	7.9 ± 0.3	7.8 ± 0.5	7.7 ± 0.5	6.6 ± 1.7
Apgar 5 min	9.1 ± 0.4	8.7 ± 0.6	8.6 ± 0.6	8.7 ± 0.5	7.7 ± 1.3

UAPI—uterine artery pulsatility index, UMAPI—fetal umbilical artery pulsatility index, MCAPI—fetal middle cerebral artery pulsatility index, Pu—proteinuria, LDH—lactate dehydrogenase, IUGR—intrauterine growth restriction.

**Table 2 diagnostics-10-01039-t002:** Number and origin of substantially represented peptides in urine of pregnant women (CH—chronic hypertension; GH—gestational hypertension; Mpe—mild preeclampsia; sPE—severe preeclamsia).

Originating Protein	Number of Peptides	Number of Samples ^a^	Diagnostic Groups ^b,c^
COL1A1	196	127	All groups
COL3A1	80	127	All groups
COL1A2	10	110	All groups
UMOD	10	102	All groups
FGA	9	121	All groups
COL18A1	8	122	Contr, CH, GH, mPE, sPE
KISS1	4	58	CH, GH, mPE
COL4A4	3	74	Contr, CH, GH, mPE
FGB	2	90	Contr, CH, GH, mPE
COL2A1	2	74	Contr, CH, GH, mPE
EMID1	2	74	Contr, CH, GH, mPE
COL5A1	2	48	CH, GH, mPE
COL8A1	2	43	CH, GH
COL15A1	1	74	Contr, CH, GH, mPE
COL17A1	1	74	Contr, CH, GH, mPE
FXYD2	1	59	Contr, mPE
PGRMC1	1	49	CH
VGF	1	49	CH,GH
INS	1	47	CH,GH
KDM6B	1	42	sPE
PIGR	1	31	GH

^a^ Values for the most represented peptides are given. ^b^ Only diagnostic groups for which the median intensity values were above zero for at least one peptide from the corresponding protein are indicated. ^c^ Gray background indicates consistent data for PEAKS and MaxQuant search programs.

**Table 3 diagnostics-10-01039-t003:** Protein affiliation of the 62 substantial peptide groups in urine of pregnant women.

Originating Protein	Number of Samples *	Intersection in Venn Diagram Number of Peptide Groups					
		Core	Control/ CH+GH	Control/ PE	CH+GH/ PE	CH+GH	PE
Intersection		47	1	1	4	7	2
UMOD	118 (102)	1	-	-	-	-	-
KISS1	85 (58)	1	-	-	-	-	-
EMID1	103 (74)	1	-	-	-	-	-
FGA	122 (121)	1	-	-	-	-	-
FGB	95 (90)	1	-	-	-	-	-
COL2A1	106 (74)	1	-	-	-	-	-
COL8A1	93 (43)	1	-	-	-	-	-
COL15A1	92 (74)	1	-	-	-	-	-
COL17A1	93 (74)	1	-	-	-	-	-
COL3A1	127 (127)	14	1	-	-	-	-
COL1A1	127 (127)	16	-	-	1	-	-
COL18A1	125 (122)	2	-	-	1	-	-
COL1A2	114 (110)	5	-	-	-	1	-
COL4A4	106 (74)	1	-	-	-	1	-
FXYD2	59 (59)	-	-	1	-	-	-
PIGR	57 (31)	-	-	-	1	-	-
COL5A1	53 (48)	-	-	-	1	1	-
COL5A2	48 (-)	-	-	-	-	1	-
INS	50 (47)	-	-	-	-	1	-
VGF	52 (49)	-	-	-	-	1	-
PGRMC1	51 (49)	-	-	-	-	1	-
KDM6B	62 (42)	-	-	-	-	-	1
SERPINA1	47 (-)	-	-	-	-	-	1

* Given values correspond to the most represented peptide groups; values in brackets indicate the numbers corresponding to the most represented peptides originating from each protein (data from Table 2).

**Table 4 diagnostics-10-01039-t004:** Urine VIP-peptide groups mostly differentiating mPE and sPE from control, CH, and GH samples.

Peptide Group	Start-End Position	Originating Protein	Number of Samples	Other Studies
GAAGEPGKAGERGVPGPPGAVGPAGKDGEAGAQGPPGPAGPAG	587–629	COL1A1	114	
EAEDLQVGQVELGGGPGAGSLQPLALEGSLQ	57–87	INS	50	
LMIEQNTKSPLFMGKVVNPTQK	397–418	SERPINA1	47	[42,45]
ERGSPGPAGPKGSPGEAGRPGEAGLPGAKG	510–539	COL1A1	98	[44]
LLGPKGPPGPPGPPGVT	683–699	COL5A1	39	
GRDGEPGTPGNPGPPGPPGPPGPPG	150–174	COL2A1	106	
GPAGPPGPPGPPGTSGHPGSPGSPGYQGPPGEPGQAGPSGPPG	174–216	COL3A1	113	
GQPGPPGPPGPPG	1021–1033	COL4A4	49	
AGPPGRDGIPGQPGLPGPPGPPGPPGPPGLGGN	126–158	COL1A1	116	
GPQGQPGLPGPPGPPGPPGPPA	551–572	COL8A1	93	
MGVVSLGSPSGEVSHPRKT	321–339	AHSG	26	
GDQPAASGDSDDDEPPPLPRL	48–68	PGRMC1	51	
ADEAGSEADHEGTHSTKRGHAKSRPV	604–624	FGA	122	[44]
ERGEQGPAGSPGFQGLPGPAGPPGEAGKPGEQGVPGDLGAPGPSG	630–674	COL1A1	127	
VKGERGSPGGPGAAGFPGARGLPGPPGSNGNPGPPGPSGSPGKDGPPGPAG	859–909	COL3A1	125	
GERGPPGPPGRDGEDGPTGPPGPPGPPGPPGLGGNFA	42–78	COL1A2	100	
LTGPIGPPGPAGAPGDKGESGPSGPAGPTG	765–794	COL1A1	110	[44]
PPPPPPPPPPPP	252–263	KDM6B	62	
LDGAKGDAGPAGPKGEPGSPGENGAPGQMGPRG	273–305	COL1A1	114	
PGERGPPGPPGPPGPPGPPAP	241–260	EMID1	103	
LTGSPGSPGPDGKTGPPGPAGQDGRPGPPGPPGA	540–573	COL1A1	127	
APGDRGEPGPPGPAG	798–812	COL1A1	126	

The results of ROC-analysis (Figure 3C,D) suggest that the obtained VIP-peptide groups can be considered as a differentiating panel for PE diagnosis. It is noteworthy that the relative content of common peptide loci may have an essential PE differentiating capacity, in addition to the presence of particular marker peptides.

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
