# Peer review of "Differential Diagnosis of Preeclampsia Based on Urine Peptidome Features Revealed by High Resolution Mass Spectrometry"

_diagnostics, 2020, doi:10.3390/diagnostics10121039_

Round 1

Reviewer 1 Report

In the study authors performed the urine proteomic assay that could be a tool for diagnostics and appropriate management. The mass spectrometry analysis was applied to analyse 127 urine samples of pregnant women with various hypertensive complications. A great study has been done, but I would like to make a few essential remarks for major revision of the manuscript:

  1. The introduction should be shortened and more focused on the problem of the study. In the present case, the introduction is very long, and does not reveal the problem that the authors address in the study.

  1. In the section Materials and Methods the preparation of urine samples should be described in detail (lines 173-177).

  1. Mass spectrometry results showed several proteins characteristic of the general PE group. However, these proteins - collagen, fibrinogen, insulin, alpha-antitrypsin, glycoproteins are the most abundant proteins and are involved in different processes. The authors themselves state that peptides are present in most samples from all diagnostic groups. Therefore, further approval is still needed. It would be appropriate to confirm the mass spectrometry results by analysis of western blot analysis and/or gene expression of identified peptides/originating proteins.

  1. Most interesting detected as possible PE marker could be KDM6B. Nerveless authors, stated that KDM6B peptides demonstrate low specificity. The authors could expand the discussion on the links between PE cases and epigenetic regulation.

Author Response

Dear Reviewer, thank you for your positive view on our study and essential remarks! Please find below our answers step by step.

1) The introduction should be shortened and more focused on the problem of the study. In the present case, the introduction is very long, and does not reveal the problem that the authors address in the study.

 Answer 1:

Following your request, the ‘Introduction’ was essentially shortened and focused on the problem of the study.

2) In the section Materials and Methods the preparation of urine samples should be described in detail (lines 173-177).

  Answer 2:

The more detailed description of urine peptides preparation is added.

3) Mass spectrometry results showed several proteins characteristic of the general PE group. However, these proteins - collagen, fibrinogen, insulin, alpha-antitrypsin, glycoproteins are the most abundant proteins and are involved in different processes. The authors themselves state that peptides are present in most samples from all diagnostic groups. Therefore, further approval is still needed. It would be appropriate to confirm the mass spectrometry results by analysis of western blot analysis and/or gene expression of identified peptides/originating proteins.

   Answer 3: Thank you for this comment! We agree that validation of the identified peptides/originating proteins is essential and in fact, it was done in our and other previous PE studies. For example, SERPINA1 protein was identified also on proteomic level in urine Congo red-aggregates (I.A. Buhimschi et al; V.A. Sergeeva). And despite of SERPINA1 peptides demonstrated substantial presence in other non-PE groups, nevertheless C-terminus peptides cluster(397-418) was shown to be the only one substantially represented in PE-samples.

In this study we are analyzing endogenous peptides and demonstrate their sequence heterogeneity not only between different groups but even in the frame of one group. Due to such a heterogeneity and sequence difference, validation of endogenous peptides is not easy using complementary quantitative methods (western blot or MRM). For this reason sequence alignment and clustering of peptides is also essential in this study. In seems that additional hydrolysis may be benefit for further peptide panel generation (from endogenous peptides) and their validation by quantitative proteomics such as MRM.

4) Most interesting detected as possible PE marker could be KDM6B. Nerveless authors, stated that KDM6B peptides demonstrate low specificity. The authors could expand the discussion on the links between PE cases and epigenetic regulation.

  Answer 4:

As recommended, we added discussion of KDM6B peptides:

‘Unlike SERPINA1 peptides, poly-P peptides are not absolutely specific markers; nevertheless, their relative content is definitely an essential indicator, first identified in this study.’

‘the appearance of KDM6B peptides may reflect the epigenetic regulation of compensatory mechanisms, which, in particular, may lead to changes in the expression level of different SERPINs [16]’

Reviewer 2 Report

This is an interesting paper describing the identification of a panel of urinary peptides that may have potential in the differential diagnosis of preeclampsia.

There are a number of items that should be addressed;

Could the authors describe why they used the same samples for the discovery work and the validation using the developed model? Usually an independent sample set is used for testing in such models? This needs to be addressed by the authors.

Why was an LTQ-FT MS instrument used to generate the peptidomic data and not the Q-Exactive instrument (which would be much faster and sensitive and should identify more peptides?)?

Line 199 – could the ‘additional’ 95 samples mentioned here be clarified, as I am not sure what they refer to?

A number of grammatical errors are present in the manuscript, please re-check manuscript and some examples are provided below;

Title – what is ‘Featches’?

Abstract, line 21- change ‘The high-resolution .. ‘to ‘High resolution ..’

Line 85, change ‘as well as may affect membrane receptors ..’ to ‘as well as affecting membrane receptors ..’

Line 89, change ‘role’ to ‘roles’

Line 94, ‘The produced by trophoblasts PlGF ..’ – please rewrite this sentence

Line 115, change ‘the previous urine peptidome/proteome studies and provided the most of the current information’ to ‘previous urine peptidome/proteome studies and provided most of the current information’

Line 255, change ‘in accord’ to ‘in accordance’

Line 337, change ‘Totally, of 50 PE-samples C-terminal peptides ..’ to ‘From a total of 50 PE-samples C-terminal peptides ..’

Line 426, rewrite this sentence as it reads poorly ‘Regarding the immediate PE markers, obtained results reinforce I.A. Buhimschi et al. SERPINA1 ..’

Author Response

This is an interesting paper describing the identification of a panel of urinary peptides that may have potential in the differential diagnosis of preeclampsia.

There are a number of items that should be addressed;

Answer: Dear Reviewer, thank you for your positive view on our study and essential remarks! Please find below our answers step by step.

Could the authors describe why they used the same samples for the discovery work and the validation using the developed model? Usually an independent sample set is used for testing in such models? This needs to be addressed by the authors.

Answer: Thank you for this comment! In this study we used cross-validation (https://en.wikipedia.org/wiki/Cross-validation_(statistics). ). When evaluating a model, the available data were divided into several parts. The quality of statistical models was estimated by R2 (fraction of data that the model can explain using the latent variables) and Q2 (fraction of data predicted by the model according to the cross validation) parameters (as described in 2.5. Data Analysis).

 Why was an LTQ-FT MS instrument used to generate the peptidomic data and not the Q-Exactive instrument (which would be much faster and sensitive and should identify more peptides?)?

  Answer: Thank you for this comment! We start this study in our Moscow lab and have access only to LTQFT MS. We believe this hybrid instrument have several benefits for the study of endogenous urine peptides. First of all this is high resolution of LTQFT which is essential for endogenous peptides accurate mass measurement as they are heavier than tryptic (up tot 10kDa). Also there is a possibility for MS/MS data acquisition in LTQ mode which is fast and sensitive for further sequencing and PTMs search. In this study we used Q-Exactive on the last stage of our project at the laboratory of our Swiss collaborators (ETH, Zurcih) and only for additional proteomic analysis as described in 2.4. Urinary Proteome Data Base Development. We agree that Q-Exactive would be interesting to compare on peptidomic level in future!

Line 199 – could the ‘additional’ 95 samples mentioned here be clarified, as I am not sure what they refer to?

  Answer: Please excuse us for missing these details. This were the 95 samples selected from our 127 patients (section - 2.1. Patients) for additional proteomic analysis. This information was added.

A number of grammatical errors are present in the manuscript, please re-check manuscript and some examples are provided below;

Answer: Please excuse us for grammatical errors. We do our best to revise manuscript to follow this comment.

Title – what is ‘Featches’?

  Answer: Changed to ‘Features’

Abstract, line 21- change ‘The high-resolution .. ‘to ‘High resolution ..’

Answer: Changed

Line 85, change ‘as well as may affect membrane receptors ..’ to ‘as well as affecting membrane receptors ..’

Answer: This sentence was rewritten in the new version.

Line 89, change ‘role’ to ‘roles’

Answer: This sentence was rewritten in the new version.

Line 94, ‘The produced by trophoblasts PlGF ..’ – please rewrite this sentence

Answer: This sentence was rewritten as recommended in the new version.

Line 115, change ‘the previous urine peptidome/proteome studies and provided the most of the current information’ to ‘previous urine peptidome/proteome studies and provided most of the current information’

Answer: Changed

Line 255, change ‘in accord’ to ‘in accordance’

Answer: Changed

Line 337, change ‘Totally, of 50 PE-samples C-terminal peptides ..’ to ‘From a total of 50 PE-samples C-terminal peptides ..’

Answer: Changed

Line 426, rewrite this sentence as it reads poorly ‘Regarding the immediate PE markers, obtained results reinforce I.A. Buhimschi et al. SERPINA1 ..’

Answer:

Thank you! The sentence was rewritten as ‘As for direct markers of PE, the obtained results support the findings of I.A. Buhimschi et al. concerning SERPINA1’

Round 2

Reviewer 1 Report

No more comments and suggestions.

Reviewer 2 Report

Corrections are fine